# Evaluating Osteotomy Accuracy in Mandibular Reconstruction: A Preliminary Study Using Custom Cutting Guides and Virtual Reality

**DOI:** 10.3390/diseases13030081

**Published:** 2025-03-13

**Authors:** Claudia Borbon, Andrea Novaresio, Oreste Iocca, Francesca Nonis, Sandro Moos, Enrico Vezzetti, Guglielmo Ramieri, Emanuele Zavattero

**Affiliations:** 1Division of Maxillofacial Surgery, Città della Salute e della Scienza University Hospital, 10126 Torino, Italy; 2Department of Management and Production Engineering, Politecnico di Torino, 10129 Torino, Italy

**Keywords:** maxillofacial surgery, fibula free flap, virtual surgical planning, VSP, virtual reality, VR

## Abstract

Background: Mandibular reconstruction has evolved significantly since its inception in the early 1900s. Currently, the fibula free flap (FFF) is considered the gold standard for mandibular and maxillary reconstructions, particularly for extensive defects, and the introduction of Extended Reality (XR) and virtual surgical planning (VSP) is revolutionizing maxillofacial surgery. Methods: This study focuses on evaluating the accuracy of using in-house cutting guides for mandibular reconstruction with FFF supported by virtual surgical planning (VSP). Planned and intraoperative osteotomies obtained from postoperative CT scans were compared in 17 patients who met the inclusion criteria. The proposed analysis included measurements of deviation angles, thickness at the centre of gravity, and the maximum thickness of the deviation volume. Additionally, a mandibular resection coding including 12 configurations was defined to classify and analyze the precision of mandibular osteotomies and investigate systematic errors. Preoperative, planned, and postoperative models have been inserted in an interactive VR environment, VieweR, to enhance surgical planning and outcome analysis. Results: The results proved the efficiency of adopting customized cutting guides and highlighted the critical role of advanced technologies such as CAD/CAM and VR in modern maxillofacial surgery. A novel coding system including 12 possible configurations was developed to classify and analyze the precision of mandibular osteotomies. This system considers (1) the position of the cutting blade relative to the cutting plane of the mandibular guide; (2) the position of the intersection axis between the planned and intraoperative osteotomy relative to the mandible; (3) the direction of rotation of the intraoperative osteotomy plane around the intersection axis from the upper view of the model. Conclusions: This study demonstrates the accuracy and reliability of in-house cutting guides for mandibular reconstruction using fibula free flaps (FFF) supported by virtual surgical planning (VSP). The comparison between planned and intraoperative osteotomies confirmed the precision of this approach, with minimal deviations observed. These findings highlight the critical role of CAD/CAM and XR technologies in modern maxillofacial surgery, offering improved surgical precision and optimizing patient outcomes.

## 1. Introduction

Functional and esthetic reconstruction of mandibular defects has been a challenge for surgeons for many years. A successful mandibular reconstruction includes a healed wound, restoration of facial contours, and facilitation of speech, swallowing, and breathing. The accuracy of the reconstruction has a significant impact on cosmetic and functional outcomes [1]. To date, mandibular reconstruction with a fibula free flap (FFF) is considered to be the best reconstructive technique for addressing mandibular or maxillary defects, whether said defects are due to benign or malignant pathologies, involving a huge bone sacrifice and potentially muscular and skin components [2].

For patients with malignant or benign tumours who need to undergo segmental resection of the mandible and subsequent reconstruction with a titanium plate in combination with a fibula free flap, precise determination of resection planes is critical. Indeed, they allow precise placement of bone segments, restoring the continuity of the mandible and associated function and esthetics and enabling the contour of the neo-mandible to match the native resected one [3]. At first, the success of mandibular osteotomies and the harvesting of correctly sized flaps depended exclusively on the skill and experience of the surgeon. However, over the past two decades, the introduction and the adoption of virtual surgical planning (VSP) have contributed to a significant evolution in oral and maxillofacial surgery [4]. Technological progress and the adoption of computer-aided design/computer-aided manufacturing (CAD/CAM) technology in the surgical field [2,5,6] have made it possible to reduce surgical times, improve the precision of bone resection thanks to accurate preoperative planning, and reduce the number of surgical interventions with adverse outcomes [4,7,8,9]. Furthermore, it became possible to obtain anatomical replicas, create customized devices and model standard fixation plates before surgery.

To report the position and orientation of the resection planes defined during virtual programming in the operating room, cutting guides customized to the patient’s anatomy are designed and manufactured using CAD techniques and additive manufacturing techniques, respectively. As reported in the literature, these customized devices enable the surgeon to perform the surgical procedure more accurately while significantly shortening the operating time [3,10].

The current study investigates the deviation between the planned osteotomy, defined during the VSP, and the intraoperative one in 17 patients who underwent mandibular reconstruction. The results of the analysis of the osteotomies report the angle between the planned and the intraoperative osteotomy, the thickness in the centre of gravity, and the maximum thickness of the deviation volume. For a complete evaluation of mandibular resections, a coding system has been introduced, summarizing (1) the position of the cutting blade relative to the mandibular guide’s cutting plane; (2) the position of the intersection axis between the planned and intraoperative osteotomy; and (3) the rotation direction of the intraoperative osteotomy plane.

Recent advancements in Virtual and Augmented Reality (VR and AR) technologies have enhanced VSP by improving preoperative planning, intraoperative guidance, and postoperative analysis [11,12,13,14]. The interactive viewer presented in this study exemplifies VR integration in postoperative analysis. Indeed, it allows surgeons to interact with virtual 3D models; visualize osteotomies overlaid on the planned model, ensuring that the cutting guides are correctly positioned and that the defined plan is accurately followed; and analyze surgical outcomes, making it easier to understand deviations and improve future procedures.

The aim of this study is (1) to assess the accuracy of using in-house cutting guides for mandibular reconstruction with fibular free flap by comparing the planned resection planes defined during the VSP to the intraoperative osteotomies, obtained from the postoperative CT; (2) to present a virtual reality environment for pre- and postoperative analysis.

## 2. Materials and Method

This study included 17 patients who underwent partial continuity resection of the mandible from January 2018 to September 2023 at the Department of Maxillofacial Surgery of the University of Torino. The exclusion criteria were (1) hemimandibulectomy and (2) incomplete clinical and radiological records, while the inclusion criteria were as follows:

Primary mandibular reconstruction with free fibula graft in patients with any disease or pathological condition requiring partial continuity resection of the mandible;

Homogenous virtual surgical planning and surgical procedures with the use of in-house 3D printed mandibular and fibular cutting guides;

Pre- and postoperative complete clinical and imaging records;

Pre- and postoperative spiral multislice computed tomography (CT) (0.625 to 1.0 mm slice thickness);

Informed consent signed by the patient.

The data collected included patient demographics, diagnosis, medical records, operative reports, imaging studies, discharge papers, histopathology reports, and follow-up reports. Data were collected one month before surgery (T0), one week after surgery (T1), and six months after surgery (T2). The study was conducted in accordance with the principles of the Declaration of Helsinki.

### 2.1. Virtual Surgical Planning

All patients underwent a preoperative diagnostic CT scan at T0 and a postoperative CT scan at T2. For each patient, two 3D anatomical models consisting of the mandible, lower dental arch, upper dental arch, and upper skull were created via medical image segmentation techniques from DICOM files. Three-dimensional medical image segmentation software was used to convert DICOM files of the patient’s mandible and donor site to 3D anatomical models in .STL format. Then, the .STL files were imported into PROPLAN^TM^ (Materialise), a 3D planning software intended for craniomaxillofacial (CMF) purposes, such as planning orthognathic procedures, mandible and midface reconstructions, osteotomies, and bone fragment repositioning. Using a dedicated tool, surgeons and engineers collaborated during the virtual surgical planning (VSP) to define the position and orientation of the mandibular osteotomies and delimit the portion to be removed on the preoperative 3D model. Additionally, the number of fibula segments and virtual osteotomies required to shape a neo-mandible were determined. Once these preliminary steps were complete, the STL file of the patient’s fibula was imported to the bony defect, and its position was fine-tuned to faithfully reproduce the contouring considering the number of segments.

To transfer this information to the operating room, cutting guides customized to the patient’s anatomy were designed to fit around the mandible and guide the saw to the planned resection planes. The cutting guide is made up of two elements: the shell and the cutting surface. The shell, designed on the patient’s jaw, allows the surgeon to quickly and uniquely position the cutting guide on the bone. Next, the cutting tool blade is brought against the cutting surface, and the osteotomy is performed. The same procedure was used for the fibula-cutting guide.

The cutting guides were designed in SolidWorks^®^ (Dassault Systèmes, Paris, France), a 3D parametric design software, and after the surgeon approved them, considering clinical aspects of intraoperative feasibility, they were exported for 3D printing. In our in-hospital 3D laboratory, the printing was carried out using the stereolithography (SLA) Form3B+ (Formlabs) with biocompatible and sterilizable resin.

### 2.2. Surgical Procedure

Resections and reconstructions were performed with a two-team approach by maxillofacial surgeons. All patients underwent a high-resolution multislice CT scan of the head and neck and a CT angiographic study of the lower leg (1.0 mm slice thickness).

Following the virtual surgical planning carried out with a team of engineers, the mandible and fibula cutting guides were used to transfer the virtual plans to the operating room. Cutting guides were used to remove the tumour or diseased bone and harvest the correct portion of fibula using a reciprocating saw or piezoelectric saw; the repositioning guide was used to transfer the planned virtual alignment of the fibula, which was then fixed using a pre-contoured plate (Synthes, Solothurn, Switzerland). Blood circulation of the fibula was established by anastomosis of the peroneal vessels with the recipient vessels in the neck.

### 2.3. Intraoperative Mandibular Resection

The planned osteotomy is represented by a plane and its position and orientation are predetermined; on the other hand, the intraoperative osteotomy must be defined. Depending on the complexity of identifying the resection margin, the surgeon defines 1 to 30 points on successive axial planes that are crucial for re-localizing the resection plane on the postoperative CT scan at T2 (Figure 1). Each point is then inserted in axial view, verifying its correctness in coronal and sagittal views. Once the coordinates of each point were defined, the equation of the interpolating plane, corresponding to the intraoperative osteotomy, was obtained via a least squares approximation algorithm using the Matlab software (Mathworks^®^).

Figure 2 shows the points defined by the surgeon for identifying the mandibular body and the mandibular ramus osteotomies in two clinical cases using the related postoperative 3D model.

### 2.4. Mandibular Resection Coding

To compare the planned osteotomy to the ones performed in the operating room, the following conditions have been considered for a complete evaluation of mandibular resections:The position of the blade of the cutting instrument with respect to the cutting plane of the mandibular guide; it can be to the right (R) or to the left (L) of the cutting plane of the mandibular guideThe position of the intersection axis between the planned and intraoperative osteotomy with respect to the mandible; it can be lingual (or internal, I), vestibular (or external, E), or intersect with the mandible (in the middle, M)The direction of rotation of the intraoperative osteotomy plane around the intersection axis, considering the upper view of the model; it can be clockwise (C) or anticlockwise (A)

The details of a mandibular body osteotomy highlighting the three regions (i.e., internal, external, or in the middle) where it is possible to identify the intersection axis between the planned and the actual resection plane is reported in Figure 3.

Figure 4 and Figure 5 show all the possible intersection combinations between the planned and the actual osteotomy when the blade is on the left or the right of the cutting plane of the guide, respectively. Each configuration is, therefore, summarized by a string containing three codes, e.g., “L_I_C” corresponds to the condition of the blade to the left of the cutting plane, internal intersection, and clockwise rotation between the planned and the defined osteotomy on the postoperative CT. To facilitate the reading and analysis of the data, the strings were renamed with an alphanumeric code consisting of a letter and a number according to Table 1.

### 2.5. Alignment

The planning mandible was superimposed on the postoperative mandible through a surface-based alignment algorithm (3DSlicer). To obtain a reliable alignment, only the external surfaces of the mandible that had not been altered by the surgery were considered for the two models.

To minimize errors, two different registrations between the planned and the postoperative models have been realized using different regions of interest (ROIs), once for each osteotomy. As shown in Figure 6, the first ROI, consisting of the right mandibular ramus and body, was used to evaluate the discrepancy between the planned and the intraoperative mandibular body osteotomy, while a second ROI, consisting of the left mandibular ramus, was used to evaluate the discrepancy between the planned and the intraoperative mandibular ramus osteotomy. The definition of ROIs has been evaluated for each patient.

The identification of the deviation volume between the planned and intraoperative resection planes to conduct a thickness analysis consists of five main steps:Positioning of the intraoperative resection plane on the 3D model of the postoperative mandible (Figure 7a).Double alignment, through a surface-based alignment algorithm, of the two mandibles (i.e., pre- and post-surgery).Transfer of the intraoperative resection plane onto the pre-surgery mandible (Figure 7b).Identification of the deviation volume (Figure 7c).Analysis of the thickness of the deviation volume using the colorimetric map (Figure 7d)

The maximum thickness of the deviation volume and the thickness in the centre of gravity were calculated for each osteotomy.

## 3. Virtual Reality

Virtual Reality (VR) technology has become increasingly popular in recent years, with applications in different fields such as gaming, education [15], cultural heritage [16,17] and surgery [18]. In healthcare and surgical technology, the integration of virtual reality into virtual surgical planning represents a significant advancement, offering surgeons new tools to improve patient outcomes through enhanced preoperative planning, intraoperative guidance and postoperative assessment [19,20,21]. VR, due to the benefits of visuospatial vision, bimanual interaction, and full immersion, is seeing increasing use for segmentation purposes [19,21] and in the overall preoperative phase, leading to improved planning and decision-making processes [20]. When used for postoperative analysis, VR is a useful tool for comparing the actual surgical outcomes after surgery with those of the preoperative plan, offering several advantages and benefits [22]. Surgeons can analyze the efficacy of the procedure, assess the accuracy of their preoperative predictions, and identify areas for improvement in future surgeries.

In this study, we designed a VR environment to (1) assess the position of the mandibular cutting guide by the surgeon (preoperative analysis) and (2) compare the planned mandibular osteotomies with the actual outcome (postoperative analysis) in a detailed and immersive manner. By overlaying preoperative onto postoperative 3D models, surgeons can evaluate the efficacy of the surgical procedure, such as tumour removal, fibula flap placement, or correct position and orientation of the cutting tool blade.

A virtual reality viewer for pre- and postoperative analysis, called VieweR, was developed in Unity (version 2022.3.23f1), and the Oculus Pro headset has been used to render it. The interface consists of a menu through which the user (i.e., the surgeon) can manage the display of models, along with a table where interactive virtual models are positioned for user interaction.

For each patient, the user can view and interact with the following models:The preoperative model: a 3D model obtained from the preoperative TC.The postoperative model: a 3D model obtained from the postoperative TC.The planned model: Virtual Surgery Planning with fibula segments, starting with the preoperative model.Alignment (planned/post): surface-based alignment of planned and postoperative models.Osteotomies: a planned model with the superimposition of planned and actual osteotomies.The cutting guide: a preoperative model with the cutting guide in the correct position.Skull: the upper skull obtained from the preoperative TC.Note: a postoperative model with the superimposition of planned and actual osteotomies and a canvas reporting surgery data, i.e., angle, thickness in the centre of gravity, and maximum thickness.

By interacting with the menu panel, the models can be toggled on/off one at a time for a more precise display, or all at once (“Show all” and “Clear all” buttons in Figure 8a). We have deliberately designed a simple, easy and intuitive environment, as shown in Figure 8c, so that virtual reality represents only an advantage, a valuable and effective tool, and in no way an obstacle for a non-expert audience. Indeed, the Meta XR Interaction SDK for Unity has been set to interact in an immersive manner with the virtual environment, allowing surgeons to grab objects, press buttons, and navigate user interfaces physically using their hands (Figure 8b).

As outlined in [23], having tracked hands in a virtual environment has significant benefits. First, hand tracking contributes significantly to increasing immersion, i.e., the sensation of being fully engaged and absorbed in a virtual environment, and presence, i.e., the feeling of actually being in the virtual environment [24]. Secondly, it allows for far more precise interactions than would be possible with controllers, improving the accuracy of movements. Third, it facilitates and encourages interactions with the virtual environment, although some downsides can occur, such as the lack of tactile and haptic cues during object manipulation [25]. Custom hand-grab poses have been recorded to allow for the best and most natural experience in the VieweR scenario.

## 4. Results

Between January 2018 and September 2023, 17 patients met the inclusion criteria and were included in this study. The patients, 11 females and 6 males, with an average age of 56.7 years (range 23–78), required reconstruction of the mandible for tumours with vascularized fibula free flap. Table 2 shows the defects, the number of fibula segments, the amount of bone harvested, and the measurements of the defect.

The results of the mandibular resections reported the codes of the osteotomies performed in the operating room, the angle between the planned osteotomy and the intraoperative osteotomy, the thickness in the centre of gravity, and the maximum thickness of the deviation volume. For each patient, the osteotomy of the mandibular body and the osteotomy of the mandibular ramus were analyzed, and the results are reported in Table 3 and Table 4, respectively.

Considering the mandibular body osteotomies performed on 17 patients (Table 3), the mean angle between the two planes is 6.44° (standard deviation 3.78) in a range of 11.71°, the mean thickness in the centre of gravity of the deviation volume is equal to 1.69 mm, and the mean maximum thickness is equal to 2.32 mm. Considering the mandibular ramus osteotomies performed on 17 patients (Table 4), the mean angle between the two planes is 5.61° (standard deviation 3.69) in a range of 13.72°, the mean thickness in the centre of gravity of the deviation volume is equal to 1.14 mm, and the mean maximum thickness is equal to 1.82 mm.

Table 5 shows the number of events performed on the mandibular body and the mandibular ramus, referring to the resection coding. Out of 17 osteotomies of the mandibular body, 8 are of the A2 type, and 2 are of the B3, C1, D2 and D3 types. Meanwhile, out of 17 osteotomies of the mandibular ramus, 4 are of the D3 type, 3 are of the A2 and D2 types, and 2 are of the A2 and C3 types. Moreover, the A3 code occurred once for the mandibular body, while the A1, C1, and C2 codes occurred once for the mandibular ramus.

## 5. Analysis

In this analysis, boxplots showing individual values have been used to assess and compare the shape, central tendency, and variability of sample distributions related to osteotomy procedures. Body and ramus data have been plotted on the same graph to compare the distribution based on the three key metrics: the angle between the planned and the intraoperative osteotomy (Figure 9a), the thickness at the centre of gravity (Figure 9b), and the maximum thickness of the deviation volume (Figure 9c).

Examining the boxplots, it is evident that the median values for the ramus are consistently lower than those for the body across all three conditions. This suggests that, on average, the osteotomies performed in the ramus region have been more precise than those in the body. Lower median values indicate that the typical deviation from the planned osteotomy is smaller for the ramus, reflecting higher precision in this region.

The variability of the data, indicated by the spread of the boxplots (interquartile range), differs between the two groups depending on the condition being considered:

The angle between planned and intraoperative osteotomy (Figure 9a): the ramus exhibits a larger spread compared to the body, signifying greater variability in the angle deviations within the ramus group. This could imply that while the median precision is higher for the ramus, the consistency of the surgical outcome is more variable.

The thickness at the centre of gravity (Figure 9b) and Maximum Thickness of Deviation Volume (Figure 9c): conversely, the body shows a larger spread in both these thickness measurements compared to the ramus. This indicates that there is more variability in the thickness deviations for the body, suggesting that the deviations in thickness are more consistently controlled in the ramus.

In addition, two outliers have been identified in the ramus group, i.e., 3.72 in the thickness at the centre of gravity and 4.42 in the maximum thickness of the deviation volume. These outliers suggest that, in a few instances, the deviations in the ramus were significantly higher than the typical values observed. These could be due to exceptional cases or errors during the procedure.

## 6. Discussion and Conclusions

Since 1990, and more frequently in recent years, surgeons have begun adopting CAD/CAM systems to aid in the planning and execution of complex surgical operations. As the name suggests, CAD/CAM software is used for the design and manufacture of prototypes or finished products, such as anatomical replicas or surgical devices. This process often involves collaboration and interaction with industrial companies prone to adopting these up-to-date technologies. The main disadvantages of adopting this system are costs, delivery times, and communication difficulties between the medical team and the industrial partner company. To overcome these problems, the first in-house laboratories were born, establishing close collaboration between surgeons and engineers. The use of low-cost 3D printers, thanks to technological development, particularly FDM (fused-deposition modelling) or SLA (stereolithography), significantly contributed to the process. The choice of printing technology and material is closely related to the application, with relative advantages and disadvantages regarding precision, resistance, and properties (biocompatible, sterilizable, transparent material). Among the numerous benefits of the CAD/CAM method, on the other hand, it is essential to mention the reduction in the operation time and the increase in the effectiveness and precision of the surgical result compared to conventional methods [26]. These advantages are maintained and expanded with an in-house approach, where the surgeon has the opportunity to work and collaborate with a group of engineers to create customized devices for the patient, taking into consideration the surgeon’s habits and the tools used during the surgery, allowing for improved cutting guide design with reduced interference in the surgical field.

Although a previous work focused on same the topic of this paper, this is the first study to focus on a comprehensive analysis of mandibular osteotomies in 17 patients who underwent fibula free flap reconstruction using custom cutting guides to enhance surgical precision. The evaluation focused on several metrics to determine the accuracy and suitability of the surgical outcomes, including the angle of incidence between the planned and actual osteotomies, the maximum thickness, and the thickness at the centre of gravity of the deviation volume. For each patient, preoperative and postoperative 3D models were obtained from CT scans and aligned through a double surface-based alignment algorithm.

A novel coding system including 12 possible configurations was developed to classify and analyze the precision of mandibular osteotomies. This system considers (1) the position of the cutting blade relative to the cutting plane of the mandibular guide; (2) the position of the intersection axis between the planned and intraoperative osteotomy relative to the mandible; (3) the direction of rotation of the intraoperative osteotomy plane around the intersection axis from the upper view of the model. This coding system, which can be used and extended to other studies, captures crucial information in an alphanumeric string, enabling the detection of systematic surgical errors and identifying common patterns. Common patterns were identified, such as case A2 being most frequent when the blade was on the left, and cases D2 and D3 being most frequent when the blade was on the right.

As a statistical analysis, boxplots were employed to compare the precision between the mandibular body and the mandibular ramus osteotomies. The median values for the ramus were lower than those for the body across all three measured conditions (angle of incidence, centre of gravity thickness, and maximum thickness), indicating that, on average, osteotomies in the ramus region were more precise. Regarding the variability of collected data, for the angle of incidence, the ramus showed greater variability compared to the body, suggesting less consistency despite the higher average precision. On the other hand, for the thickness measurements (both centre of gravity and maximum thickness), the body displayed greater variability, indicating less controlled deviations in thickness compared to the ramus.

Finally, to enhance the surgical planning and outcome analysis, virtual reality (VR) was integrated into the virtual surgical planning (VSP). The entire workflow, starting with virtual planning, can be considered a valuable learning tool. Indeed, it is an effective way to teach technical and non-technical skills in a safe environment. The virtual model obtained from TC scans allows junior surgeons to easily and efficiently simulate the surgery following virtual guidance and share the decision-making process and indications for surgery with senior surgeons during VSP. In this study, an interactive virtual viewer called VieweR was developed and presented to allow junior and senior surgeons to visualize and interact with preoperative, postoperative, and planned 3D models, as well as the upper skull, the osteotomies overlaid on the planned model, and the cutting guide in the correct position. Therefore, VR can be effectively integrated into the surgical procedure to visualize postoperative results intuitively, interactively, and in an immersive manner, enhancing the understanding of surgical outcomes and allowing surgeons to examine the effectiveness of the surgery in detail and learn from previous experiences and procedures. As future developments, combining the visualization capabilities of VR with a training phase is envisaged, thus integrating virtual reality into both the preoperative planning and postoperative review phases. Moreover, advancements in real-time VR technology could provide surgeons with live feedback during surgery, enhancing precision and reducing errors.

## Figures and Tables

**Figure 1 diseases-13-00081-f001:**
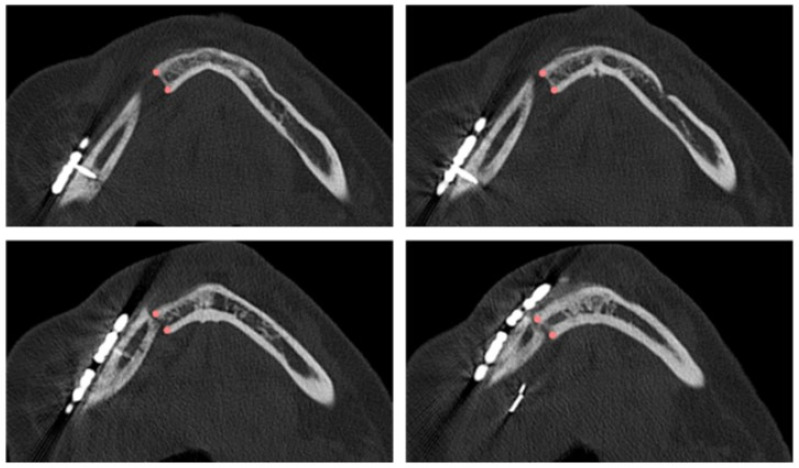
Points defined by the surgeon in four axial views of the postoperative CT scan for identifying the intraoperative resection plane.

**Figure 2 diseases-13-00081-f002:**
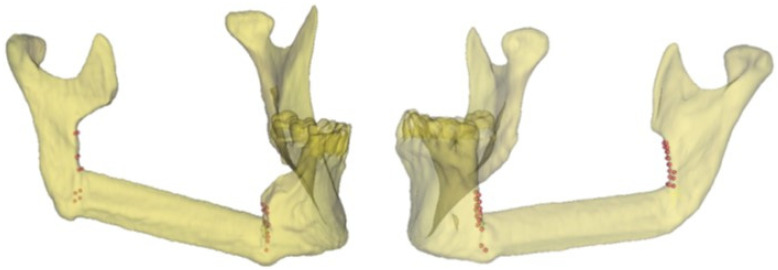
Representative points of the intraoperative resection plans (body and ramus osteotomies) in two clinical cases and the related 3D models.

**Figure 3 diseases-13-00081-f003:**
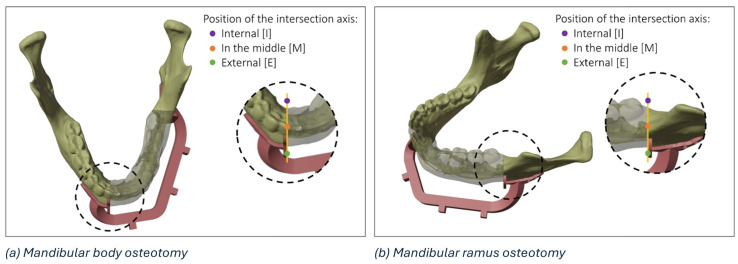
Details of the possible regions of intersection (internal, in the middle, and external) between the planned and the actual resection plane in the mandibular body (**a**) and mandibular ramus (**b**).

**Figure 4 diseases-13-00081-f004:**
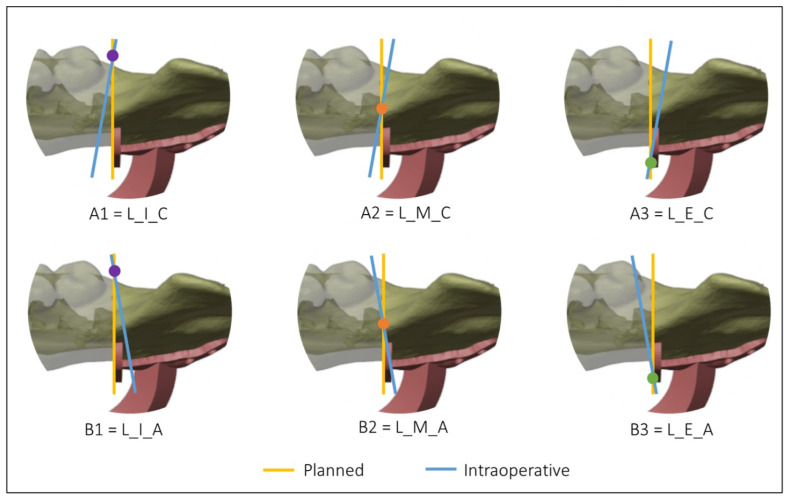
Intersections between the planned and the intraoperative osteotomy, represented by the yellow and the blue line, respectively, when the blade is to the left of the cutting guide.

**Figure 5 diseases-13-00081-f005:**
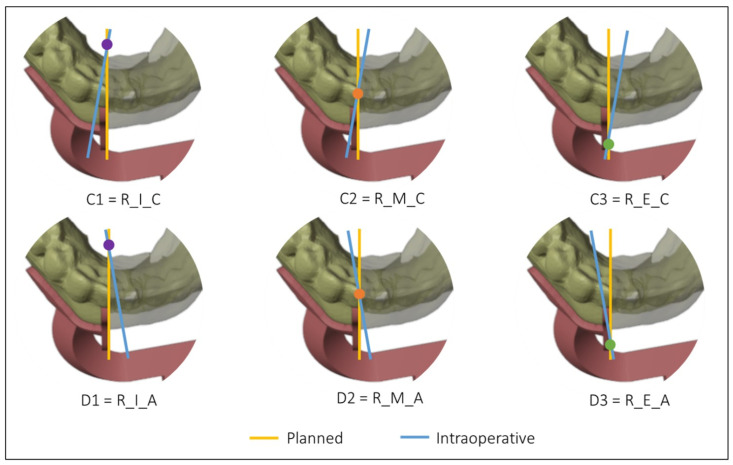
Intersections between the planned and the intraoperative osteotomy, represented by the yellow and the blue line, respectively, when the blade is to the right of the cutting guide.

**Figure 6 diseases-13-00081-f006:**
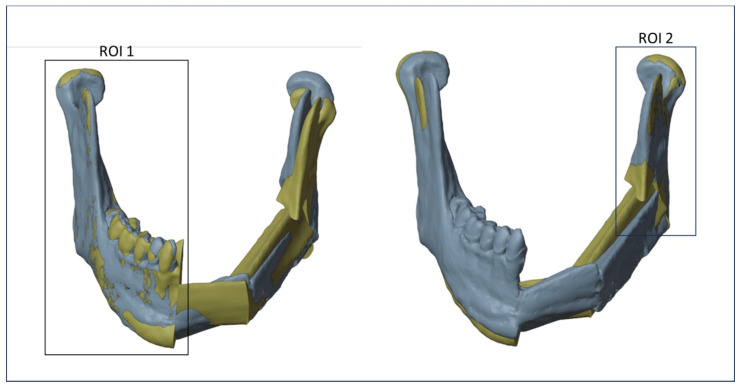
Regions of interest (ROIs) considered to align the planned and the postoperative models.

**Figure 7 diseases-13-00081-f007:**
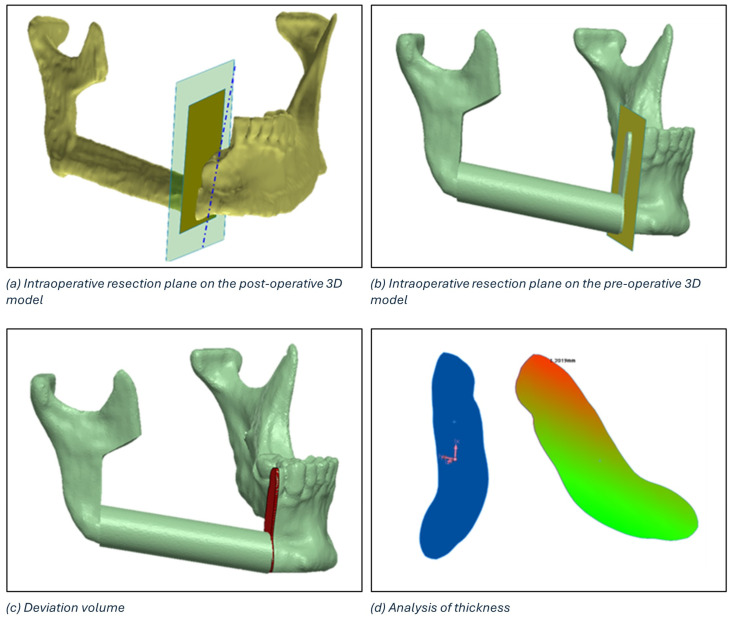
Definition of the deviation volume figure and analysis of thickness using the colorimetric map.

**Figure 8 diseases-13-00081-f008:**
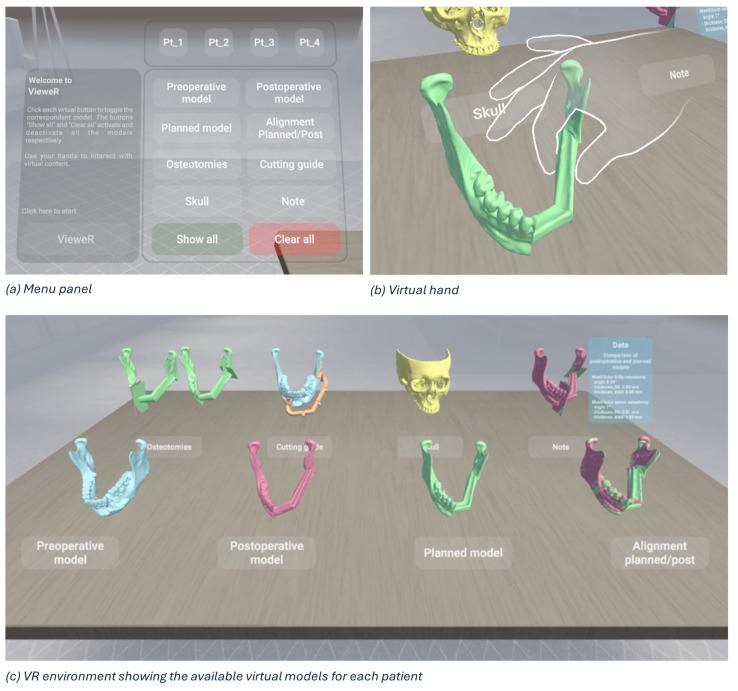
Overview of VieweR, an interactive VR environment for viewing surgical outcomes.

**Figure 9 diseases-13-00081-f009:**
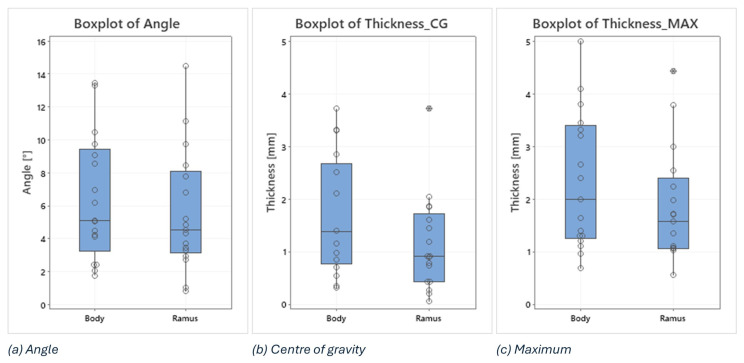
Boxplots of angle, thickness in the centre of gravity and maximum thickness of the deviation volume values, comparing body and ramus osteotomies.

**Table 1 diseases-13-00081-t001:** Simplified codes identifying the type of intraoperative mandibular resection.

Blade to the Left of the Cutting Guide	Blade to the Right of the Cutting Guide
A1 = L_I_C	C1 = R_I_C
A2 = L_M_C	C2 = R_M_C
A3 = L_E_C	C3 = R_E_C
B1 = L_I_A	D1 = R_I_A
B2 = L_M_A	D2 = R_M_A
B3 = L_E_A	D3 = R_E_A

**Table 2 diseases-13-00081-t002:** Morphological and functional analysis showing the size of the defect, the number of fibula segments required to perform the reconstruction, and the type of resection (i.e., ramus, body, and symphysis).

Mandibular Reconstruction with Vascularized FFF			
Pt	Bone Defect [cm]	N. Fibula Segments	Resection
1	6.0	1	Body and ramus
2	9.0	2	Body and ramus
3	8.5	2	Body and ramus
4	7.0	1	Body
5	6.0	1	Body and ramus
6	9.5	2	Body and ramus
7	6.0	1	Body
8	5.5	1	Body
9	8.0	2	Symphysis and body
10	7.5	2	Symphysis and body
11	6.0	1	Body
12	8.0	2	Symphysis and body
13	8.5	2	Symphysis and body
14	6.5	2	Body
15	5.5	1	Body and ramus
16	6.5	1	Body and ramus
17	9.0	2	Symphysis and body

**Table 3 diseases-13-00081-t003:** Results of the analysis of the osteotomies performed on the mandibular body, where Pt refers to the number of patients, Code refers to the mandibular resection coding, Angle refers to the angle between the planned and the intraoperative osteotomy, Thickness_CG refers to the thickness in the centre of gravity, and Thickness_MAX refers to the maximum thickness of the deviation volume.

Mandibular Body Osteotomies				
Pt	Code	Angle [°]	Thickness_CG [mm]	Thickness_MAX [mm]
1	A2	5.04	0.31	0.68
2	A2	6.95	0.54	0.96
3	A3	4.21	1.39	1.99
4	A2	6.17	0.98	1.30
5	B3	2.40	1.16	1.20
6	D2	4.50	0.34	1.10
7	A2	5.10	0.70	1.40
8	D3	9.10	1.40	2.40
9	A2	10.50	2.50	3.20
10	D3	8.57	2.85	3.80
11	A2	4.10	0.85	1.30
12	D2	13.46	0.84	1.64
13	C1	1.75	3.30	3.45
14	A2	9.77	3.32	5.00
15	C1	2.48	2.10	2.65
16	B3	2.06	3.72	4.09
17	A2	13.33	2.49	3.32
Mean		6.44	1.69	2.32
StDev		3.78	1.13	1.29
Minimum		1.75	0.31	0.68
Maximum		13.46	3.72	5.00
Range		11.71	3.41	4.32

**Table 4 diseases-13-00081-t004:** Analysis results of the osteotomies performed on the mandibular ramus, where Pt refers to the number of patients, Code refers to the mandibular resection coding, Angle refers to the angle between the planned and the intraoperative osteotomy, Thickness_CG refers to the thickness in the centre of gravity, and Thickness_MAX refers to the maximum thickness of the deviation volume.

Mandibular Ramus Osteotomies				
Pt	Code	Angle [°]	Thickness_CG [mm]	Thickness_MAX [mm]
1	D3	3.72	0.74	1.05
2	C3	8.44	1.85	2.54
3	D3	9.76	2.04	3.78
4	D3	3.45	0.78	1.35
5	C3	2.70	0.26	1.34
6	A1	5.20	0.90	1.70
7	C2	7.80	0.05	1.10
8	A2	14.50	1.60	3.00
9	D3	4.56	0.43	1.03
10	A2	1.00	0.91	1.07
11	D2	4.86	0.20	0.55
12	A2	0.79	0.43	0.56
13	B1	3.31	1.44	1.97
14	C1	4.31	1.18	1.58
15	B1	6.80	1.87	2.24
16	D2	2.95	3.72	4.42
17	D2	11.14	0.91	1.72
Mean		5.61	1.14	1.82
StDev		3.69	0.91	1.08
Minimum		0.79	0.05	0.55
Maximum		14.50	3.72	4.42
Range		13.72	3.67	3.87
**Mandibular Body Osteotomies**				
**Pt**	**Code**	**Angle [°]**	**Thickness_CG [mm]**	**Thickness_MAX [mm]**
1	A2	5.04	0.31	0.68
2	A2	6.95	0.54	0.96
3	A3	4.21	1.39	1.99
4	A2	6.17	0.98	1.30
5	B3	2.40	1.16	1.20
6	D2	4.50	0.34	1.10
7	A2	5.10	0.70	1.40
8	D3	9.10	1.40	2.40
9	A2	10.50	2.50	3.20
10	D3	8.57	2.85	3.80
11	A2	4.10	0.85	1.30
12	D2	13.46	0.84	1.64
13	C1	1.75	3.30	3.45
14	A2	9.77	3.32	5.00
15	C1	2.48	2.10	2.65
16	B3	2.06	3.72	4.09
17	A2	13.33	2.49	3.32
Mean		6.44	1.69	2.32
StDev		3.78	1.13	1.29
Minimum		1.75	0.31	0.68
Maximum		13.46	3.72	5.00
Range		11.71	3.41	4.32

**Table 5 diseases-13-00081-t005:** Occurrences of resection codes associated with osteotomies performed on the mandibular body and ramus.

Resection Code			Osteotomies			
			Body		Ramus	
A1			0		1	
A2			8		3	
A3			1		0	
B1			0		2	
B2			0		0	
B3			2		0	
C1			2		1	
C2			0		1	
C3			0		2	
D1			0		0	
D2			2		3	
D3			2		4	
**Mandibular Ramus Osteotomies**						
**Pt**	**Code**	**Angle [°]**		**Thickness_CG [mm]**		**Thickness_MAX [mm]**
1	D3	3.72		0.74		1.05
2	C3	8.44		1.85		2.54
3	D3	9.76		2.04		3.78
4	D3	3.45		0.78		1.35
5	C3	2.70		0.26		1.34
6	A1	5.20		0.90		1.70
7	C2	7.80		0.05		1.10
8	A2	14.50		1.60		3.00
9	D3	4.56		0.43		1.03
10	A2	1.00		0.91		1.07
11	D2	4.86		0.20		0.55
12	A2	0.79		0.43		0.56
13	B1	3.31		1.44		1.97
14	C1	4.31		1.18		1.58
15	B1	6.80		1.87		2.24
16	D2	2.95		3.72		4.42
17	D2	11.14		0.91		1.72
Mean		5.61		1.14		1.82
StDev		3.69		0.91		1.08
Minimum		0.79		0.05		0.55
Maximum		14.50		3.72		4.42
Range		13.72		3.67		3.87
**Mandibular Body Osteotomies**						
**Pt**	**Code**	**Angle [°]**		**Thickness_CG [mm]**		**Thickness_MAX [mm]**
1	A2	5.04		0.31		0.68
2	A2	6.95		0.54		0.96
3	A3	4.21		1.39		1.99
4	A2	6.17		0.98		1.30
5	B3	2.40		1.16		1.20
6	D2	4.50		0.34		1.10
7	A2	5.10		0.70		1.40
8	D3	9.10		1.40		2.40
9	A2	10.50		2.50		3.20
10	D3	8.57		2.85		3.80
11	A2	4.10		0.85		1.30
12	D2	13.46		0.84		1.64
13	C1	1.75		3.30		3.45
14	A2	9.77		3.32		5.00
15	C1	2.48		2.10		2.65
16	B3	2.06		3.72		4.09
17	A2	13.33		2.49		3.32
Mean		6.44		1.69		2.32
StDev		3.78		1.13		1.29
Minimum		1.75		0.31		0.68
Maximum		13.46		3.72		5.00
Range		11.71		3.41		4.32

## Data Availability

Data is available upon reasonable request.

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
