# Peer review of "Evaluating Osteotomy Accuracy in Mandibular Reconstruction: A Preliminary Study Using Custom Cutting Guides and Virtual Reality"

_diseases, 2025, doi:10.3390/diseases13030081_

Round 1

Reviewer 1 Report

Comments and Suggestions for Authors

This study neglects the clinical results : what about the importance of restoring the dental occlusion ? (L 28)

What about the necessity to change the level or amplitude of resection for any reason during the peroperative phase ?

L32 : this reviewer is not sure that the reconstructive surgery with FFF after resection of bening tumors of the maxilla  is the gold standard !!

L 127 : ma-xillo

please harmonise pre-operative and postoperative in the entire manuscript : L 237, L 241, L 248, L 255....

What means TC (L 247 and L 404) : the authors should define these abreviations = CT ?? or does it mean something else ?

The authors should give at least some insights into the costs of this tool, ea. cost benefit study ??

L 259 : gra- instead of grav-

L 282 : femeles and males

L 350-L 366 : discussion about CAD/CAM should be shortened, because this part of the discussion has nothing to do here!

L 414 : envi- and not envis-

5 self citations on 26 references seem to  this reviewer too much !!

Author Response

This study neglects the clinical results : what about the importance of restoring the dental occlusion ? (L 28)

We understand that comment of the reviewer. In this study we focused on the reconstruction of the mandible and the accuracy of the osteotomy lines using custom cutting guides. The dental occlusion is an important topic but it is not the focus of this research. Undoubtedly, a proper bone restoration will allow to perform an optimal dental occlusion via the use of dental implants.

What about the necessity to change the level or amplitude of resection for any reason during the peroperative phase ?

This is an important topic but for this small set of patients it was not a necessity to change the level or amplitude of resection during the preoperative phase. In any case if it was necessary, thanks to the use of in-house instrumentation, it would not have been difficult to do it.

L32 : this reviewer is not sure that the reconstructive surgery with FFF after resection of bening tumors of the maxilla  is the gold standard !!

We understand the comment of the reviewer. We did not explain well what we meant with this phrase and we changed it accordingly to make it clear that fibula free flap is the gold standard for large defects involving the mandible or the maxilla.

L 127 : ma-xillo

Thank you for the comment

please harmonise pre-operative and postoperative in the entire manuscript : L 237, L 241, L 248, L 255....

Thank you for the comment, we changed it accordingly

What means TC (L 247 and L 404) : the authors should define these abreviations = CT ?? or does it mean something else ?

We changed TC with CT (computed tomography)

The authors should give at least some insights into the costs of this tool, ea. cost benefit study ??

We understand this comment. We provided a general cost beenfit analysis in the text comparing the cost of the in-house custom cutting guide with industrial ones. The average cost of an industrial one not in house is around 5000$ while the in-house cost taking into accoutn the personnel work and the materials is around 500$. We added this on the line 427

L 259 : gra- instead of grav-

Thank you for the comment

L 282 : femeles and males

Thank you for the comment

L 350-L 366 : discussion about CAD/CAM should be shortened, because this part of the discussion has nothing to do here!

Thank you for the comment with shortened the discussion

L 414 : envi- and not envis-

Thank you for your comment

5 self citations on 26 references seem to  this reviewer too much !!

We reduced the the numebr of citations to 4 as we deemed them very relevant to our work

Reviewer 2 Report

Comments and Suggestions for Authors

The subject of the reviewed article is "Evaluating Osteotomy Accuracy in Mandibular Reconstruction: A Preliminary Study Using Custom Cutting Guides and Virtual Reality"

The aim of the study was to evaluate the accuracy of osteotomy during mandibular reconstruction using a fibula free flap and to verify the potential of the VR environment in planning and analyzing surgical results.
The study included 17 patients who underwent mandibular reconstruction using a fibula free flap operated on between January 2018 and September 2023.
The criteria for inclusion in the study are: primary mandibular reconstruction using FFF and cutting guides, complete pre- and postoperative clinical and imaging data (CT), and patient consent to participate in the study.
The authors describe the planning and surgical protocol in detail.
A precise cutting line was planned for each procedure and a previously prepared guide was used to facilitate its execution.
The authors compared the planned osteotomy lines with those actually made during the procedure using previously performed CT and virtual reality.

The authors developed a resection coding system that took into account three key aspects:
-the position of the blade in relation to the cutting plane,
-the position of the cutting axis in relation to the mandible (internal, external, central),
-the direction of rotation of the actual osteotomy plane.

The study shows that
-The average angular deviation between the planned and actual cutting line was 6.44° for the mandibular body and 5.61° for the mandibular ramus.
-The average thickness of the deviation was 1.69 mm for the body and 1.14 mm for the ramus, while the maximum thickness of the deviation was 2.32 mm and 1.82 mm.
-The osteotomy is more precise in the mandibular ramus, but with greater variability of angles.
The article contains 9 figures, which are legible and very helpful in understanding the article and 5 tables, which are quite extensive and difficult to analyze.

The authors emphasize that the introduction of the VR environment allowed for detailed visualization of the results and interaction with 3D models. This technology facilitated the assessment of the precision of the cuts and allowed for the comparison of pre- and postoperative models.

In summary, the article is a significant contribution to the development of modern techniques in maxillofacial surgery. The use of VR and CAD/CAM technology can improve surgical accuracy and reduce deviations between planned and actual cut lines.

The possibility of using the above-mentioned technologies in planning and during surgery would certainly improve the quality and precision of the procedures performed.

However, there are several aspects of this work that the authors should address and enrich the text of the manuscript with answers to the following issues:
- Did the deviations in the precision of osteotomy have a significant impact on such aspects as: completeness of resection, plate matching, damage to teeth, vessels, nerves, etc.?
-Do similar studies exist? If so, should the results obtained by other authors be discussed? If not, indicate that the studies are pioneering.
-Discuss the issue of whether it is possible to design guides in such a way that they force a precise cutting line?
-To what extent can the presented deviations negatively affect the final results of the treatment?
-Do the authors have any specific recommendations for preventing deviations from the planned cutting lines?
-Please describe whether there have been any attempts to use VR during Maxillofacial Surgery procedures? If so, in what specific cases?
-Please familiarize the reader with the available programs that allow for performing virtual operations in Maxillofacial Surgery and what are their characteristics?

Author Response

Reviewer 2

The subject of the reviewed article is "Evaluating Osteotomy Accuracy in Mandibular Reconstruction: A Preliminary Study Using Custom Cutting Guides and Virtual Reality"

The aim of the study was to evaluate the accuracy of osteotomy during mandibular reconstruction using a fibula free flap and to verify the potential of the VR environment in planning and analyzing surgical results.

The study included 17 patients who underwent mandibular reconstruction using a fibula free flap operated on between January 2018 and September 2023.

The criteria for inclusion in the study are: primary mandibular reconstruction using FFF and cutting guides, complete pre- and postoperative clinical and imaging data (CT), and patient consent to participate in the study.

The authors describe the planning and surgical protocol in detail.

A precise cutting line was planned for each procedure and a previously prepared guide was used to facilitate its execution.

The authors compared the planned osteotomy lines with those actually made during the procedure using previously performed CT and virtual reality.

The authors developed a resection coding system that took into account three key aspects:

-the position of the blade in relation to the cutting plane,

-the position of the cutting axis in relation to the mandible (internal, external, central),

-the direction of rotation of the actual osteotomy plane.

The study shows that

-The average angular deviation between the planned and actual cutting line was 6.44° for the mandibular body and 5.61° for the mandibular ramus.

-The average thickness of the deviation was 1.69 mm for the body and 1.14 mm for the ramus, while the maximum thickness of the deviation was 2.32 mm and 1.82 mm.

-The osteotomy is more precise in the mandibular ramus, but with greater variability of angles.

The article contains 9 figures, which are legible and very helpful in understanding the article and 5 tables, which are quite extensive and difficult to analyze.

The authors emphasize that the introduction of the VR environment allowed for detailed visualization of the results and interaction with 3D models. This technology facilitated the assessment of the precision of the cuts and allowed for the comparison of pre- and postoperative models.

In summary, the article is a significant contribution to the development of modern techniques in maxillofacial surgery. The use of VR and CAD/CAM technology can improve surgical accuracy and reduce deviations between planned and actual cut lines.

The possibility of using the above-mentioned technologies in planning and during surgery would certainly improve the quality and precision of the procedures performed.

However, there are several aspects of this work that the authors should address and enrich the text of the manuscript with answers to the following issues:

- Did the deviations in the precision of osteotomy have a significant impact on such aspects as: completeness of resection, plate matching, damage to teeth, vessels, nerves, etc.?

We thank the reviewer for the comment. The precision of the osteotomy has an importance for the morphology of the reconstruction. The damage to the anatomical structures is the same regardless of the precision of the osteotomy (given that the same strucures are removed wither with custom guides or not). For this reason we focused only on the reconstruction results.

-Do similar studies exist? If so, should the results obtained by other authors be discussed? If not, indicate that the studies are pioneering.

We thank the reviewer for the comment. There is one study similar to ours that we now added to the text [ Zavattero E, Bolzoni AM et al;. Accuracy of Fibula Reconstruction Using Patient-Specific Cad/Cam Plates: A Multicenter Study on 47 Patients. Laryngoscope. 2021 Jul;131(7):E2169-E2175. doi: 10.1002/lary.29379. Epub 2021 Jan 16. PMID: 33452834]

-Discuss the issue of whether it is possible to design guides in such a way that they force a precise cutting line?

The custom made guides by definition define a precise cutting guide so they allow to position the cut exactly where it is planned

-To what extent can the presented deviations negatively affect the final results of the treatment?

The presented deviations do not negatively affect the clinical results but they allow to understand that always the same deviation is present.

-Do the authors have any specific recommendations for preventing deviations from the planned cutting lines?

We recommend to use ergonomic instruments that are compatible with the premade cutting guides. We specify this in the text.

-Please describe whether there have been any attempts to use VR during Maxillofacial Surgery procedures? If so, in what specific cases?

There have been still no attempts to use the VR in a clinical setting. We presented it just a demonstration and we plan to adopt it in clinical practice in future studies

-Please familiarize the reader with the available programs that allow for performing virtual operations in Maxillofacial Surgery and what are their characteristics?

We thank the reviewer for allowing us to specify the characteristics of the available programs and how we implemented it in the text as follow:

Digital Imaging and Communications in Medicine data of a preoperative CT scan of the patient’s mandible and donor site were transferred to a free, public software three-dimensional (3D) slicer (Brigham and Women’s Hospital, Harvard University, Boston, MA). The segmentation and the thresholding of the skull and theflap donor site were carried out. 3D stereolithography (STL) files of the models were extracted.
The STL files were imported to Blender, the free computer-aided design (CAD) software (The Blender Foundation, Amsterdam, the Netherlands) or to the professional CAD software SolidWorks (Dessault Systems,
Vélizy-Villacoublay, France). The planes for osteotomy were set using the software tools. After setting the  planes for osteotomy, the resection of the bone was performed virtually. The STL files were then imported to Meshmixer free software (Autodesk, San Rafael, CA). After defining the mandible plane thickness, the solid mandible was subtracted from this to create a thick osteotomy plane that could be fit to the outer surface of the mandible. These procedures were performed at both edges of the osteotomies. After the solids were
united, a prop was used to connect the bodies, and the cutting guide was designed. This guide was planned to include information for surgery such as the planes for osteotomies, the connection to the bone after resection, and the relationship between the remaining bones. In five cases the attached resection guide was used as external fixator. For the fibula cutting guide the same procedure was used. The virtual cutting guides for both the mandible and the fibula were exported as STL files and applied by using the 3D printer. In our in-hospital 3D
laboratory, the printing was carried out using an Ultimaker (Utrecht, the Netherlands) extended printer.

Round 2

Reviewer 1 Report

Comments and Suggestions for Authors

The authors take time to answer the questions and do the asked corrections

Reviewer 2 Report

Comments and Suggestions for Authors

The authors have made the corrections indicated in round 1. The work in its current state is suitable for printing.